# Point–Line-Aware Heterogeneous Graph Attention Network for Visual SLAM System

**Yuanfeng Lian** [1,2,*] **, Hao Sun** [2] **and Shaohua Dong** [3]

1 Beijing Key Laboratory of Petroleum Data Mining, Beijing 102249, China
2 Department of Computer Science and Technology, China University of Petroleum, Beijing 102249, China
3 Pipeline Technology and Safety Research Center, China University of Petroleum, Beijing 102249, China
* Correspondence: lianyuanfeng@cup.edu.cn

**Abstract:** Simultaneous localization and mapping (SLAM), as an important research topic in robotics, is useful but challenging to estimate robot pose and reconstruct a 3-D map of the surrounding environment. Despite recent success of several deep neural networks for visual SLAM, those methods cannot achieve robust results in complex industrial scenarios for constructing accurate and real-time maps due to the weak texture and complex geometric structure. This paper presents a novel and efficient visual SLAM system based on point–line-aware heterogeneous graph attention network, which combines points and line segments to solve the problem of the insufficient number of reliable features in traditional approaches. Firstly, a simultaneous feature extraction network is constructed based on the geometric relationships between points and points and points and lines. To further improve the efficiency and accuracy of the geometric association features of key regions, we design the point–line-aware attention module to guide the network to pay attention to the trivial features of both points and lines in images. Moreover, the network model is optimized by a transfer-aware knowledge distillation strategy to further improve the system's real-time performance. Secondly, to improve the accuracy of the point–line matching, we design a point–line heterogeneous graph attention network, which combines an edge aggregation graph attention module and a cross-heterogeneous graph iteration module to conduct learning on the intragraph and intergraph. Finally, the point–line matching process is transformed into an optimal transport problem, and a near-iterative method based on a greedy strategy is presented to solve the optimization problem. The experiments on the KITTI dataset and a self-made dataset demonstrate the better effectiveness, accuracy, and adaptability of our method than those of the state of the art in visual SLAM.

**Keywords:** visual SLAM; point–line aware; knowledge distillation; heterogeneous graph attention network

## 1. Introduction

Simultaneous localization and map construction technology, as the key to autonomous movement of robots, is widely used in unmanned driving, virtual reality, mobile robots, and other fields [1–5]. Compared with laser SLAM, vision-based SLAM has a low power consumption, low cost, miniaturization, and other advantages, and its theoretical and application value is very prominent [6]. Visual SLAM constructs a map of the surrounding environment by obtaining the plane image information of the real world through a camera. The pose state of the camera is inferred by the extracted feature information or pixel grayscale. Visual SLAM methods can be classified into four categories: feature-point-based methods [6–8], feature-line-based methods [9–12], feature-plane-based methods [12,13], and the combination of the above methods [14–20], according to the kind of features used to estimate the trajectory. The existing empirical methods cannot deal effectively with complex industrial scenarios due to occlusion, illumination, and deformation issues. In recent years, research on visual SLAM based on deep learning has attracted widespread

attention. Serra et al. [21] used deep convolution to extract a point description with the L2 norm as a similarity measure to enhance the robust matching between key points and their local image features. Since a single feature descriptor cannot generate an effective key-point detection, Shend et al. [22] proposed an end-to-end trainable matching network, RF-Net, based on a receptive field to achieve a more efficient key-point detection. In order to solve the problem of the low localization accuracy caused by the lack of key-point shape perception in the joint learning of feature detectors and descriptors, Luo et al. [23] applied a deformable convolutional network with a dense spatial transformation to enhance the dynamic receptive field and improve the ability to express local shapes. Sarlin et al. [24] proposed a graph neural network, SuperGlue, based on attention aggregation, which used the optimal transmission model for matching optimization and realized the pose estimation in both indoor and outdoor environments. Combining a convolutional network with a recurrent network, Tang et al. [25,26] proposed a geometric correspondence network (GCN), which used an end-to-end learning method to detect key points and generate descriptors for improving the accuracy of the pose estimation. Aimed at the problem of decreased localization accuracy caused by the partial occlusion of line segments, Pautrat et al. [27] used a self-supervised network for line detection and for extracting line-segment descriptors, which improved the robustness of line-segment matching. The above-mentioned feature detection algorithms based on deep learning fused multilevel features, which did not deeply explore the association and constraint relationship between point and line features.

In industrial production scenarios, there are many complex background objects such as various buildings, pipelines, production equipment, and safety signs that lack corner points or contain repeated textures. The point features in the image are not specific enough, which makes them unable to provide an accurate position estimation. In addition, the mismatch of line features greatly increases the time complexity of the computation. In general, although existing methods have achieved certain results in feature detection and matching tasks, due to the uneven light, single texture, and complex scene structure in industrial scenarios, the pose estimation is easily degraded. How to efficiently fuse point and line information to build a more stable visual SLAM system is still a difficult problem that needs further research. Our main contributions can be summarized as follows:

- To solve the problem of weak point–line extraction ability in complex scenes, a point–line synchronous geometric feature extraction network, PL-Net, is proposed. We use an optimized residual block-feature pyramid network (ORB-FPN) to extract the feature map of the input image. In the point extraction branch, based on the point-aware module, the multiscale context is aggregated to obtain features with rich receptive fields. Moreover, the edge information is used for the line extraction branch to improve the accuracy of the line-segment detection. In order to make the network lightweight, a transfer-aware knowledge distillation method is proposed to compress the model for generating the point–line feature in the extraction task.
- Targeting a high accuracy and efficiency, a heterogeneous attention graph neural network (HAGNN) is presented, which uses an edge-aggregated graph attention network (EAGAT) to iterate the vertices of the heterogeneous graph constructed from points and lines. To enhance the performance of the point–line matching, a cross-heterogeneous graph interaction (CHGI) is used for harmonizing heterogeneous information between graphs.
- By transforming the point–line matching process into an optimal transport problem, a greedy inexact proximal point method for optimal transport, GIPOT, is proposed, which calculates the optimal feature assignment matrix to find the global optimal solution for the point–line matching problem.

## 2. SLAM System Framework

The framework of the SLAM system proposed in this paper is shown in Figure 1. Firstly, a new image is input into the PL-Net network to detect key points and line segments, and the corresponding descriptors are obtained through the point–line-aware attention

module to enhance the feature expressiveness for both points and lines in images. Then, the point and line features of the two images are transferred to the point–line heterogeneous graphs, which are constructed by using the point and line features as the vertices and connecting a vertex to its neighbors within a fixed radius. Secondly, the attention network HAGNN obtains the enhanced features and inputs them into the GIPOT to generate point–line matching results and calculate the pose of the current frame. Finally, by reprojecting the features in the local map to the current frame, the projection error is calculated for the backend processing of SLAM to complete the map.

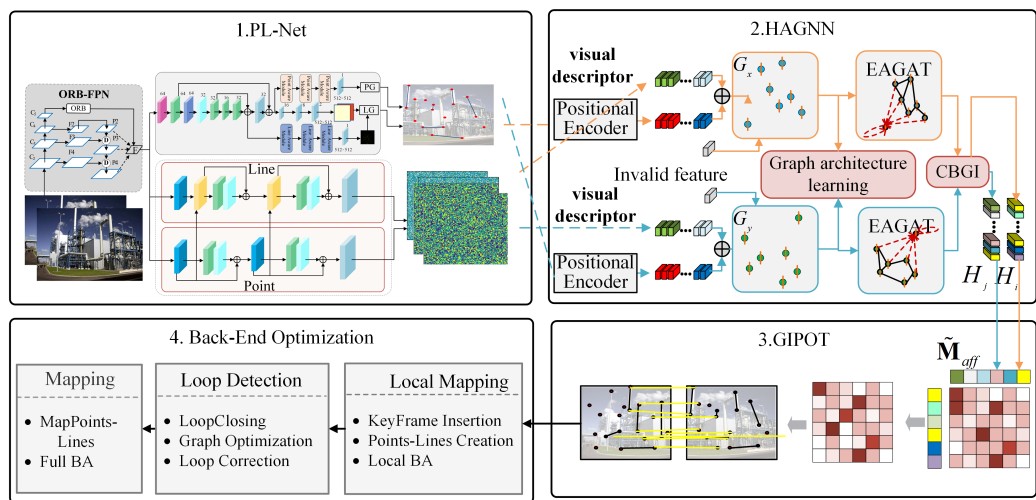

**Figure 1.** System overview. The system has four components: 1. The point–line feature extraction network (PL-Net) extracts key points, line segment and their descriptors (Section 3.1). 2. A heterogeneous graph attention network (HAGNN) is added with the positional encoding, which has $N$ EAGAT and GBGI layers (Section 3.2). 3. A greedy near iterative matching (GIPOT) module is used to match the transformed features, which computes the affinity matrix $\tilde{M}_{aff}$ and the assignment matrix. (Section 3.3). 4. The backend optimization includes local mapping, loop detection, and mapping.

## 3. Methodology

### 3.1. Point–Line Feature Extraction Network

The point–line feature extraction network PL-Net is shown in Figure 2. Firstly, the ORB-FPN module was used to extract the features of each layer for the image, and the PG module completed the key-point extraction. Then, the center point map and displacement map were generated through the branch of the line-segment perception module. Finally, the point–line descriptor was generated through convolution and upsampling operations.

### 3.1.1. ORB-FPN Module

As shown in Figure 2, an optimized residual block (ORB) is designed based on the Nesterov acceleration gradient (NAG) algorithm to enhance the expressive ability of target features [26]. Then, we have:

$$y_{k+1} = x_k + \beta(x_k - x_{k-1}) \tag{1}$$

$$x_{k+1} = y_{k+1} - \alpha \nabla f(y_{k+1}) \tag{2}$$

where $x_k$ and $y_{k+1}$ denote the output and input of the first layer of the network, and $\alpha$ and $\beta$ represent the learning rate and momentum parameters, respectively. $\nabla f(y_{k+1})$ is the gradient of the objective function $f$ at $y_{k+1}$, and $f$ is a smooth function satisfying the Lipschitz property. When the momentum parameter $\beta = 0$, the NAG algorithm is equivalent to the standard gradient descent algorithm. When $\beta > 0$, it optimizes the combination of $\alpha$ and $\beta$ to accelerate convergence.

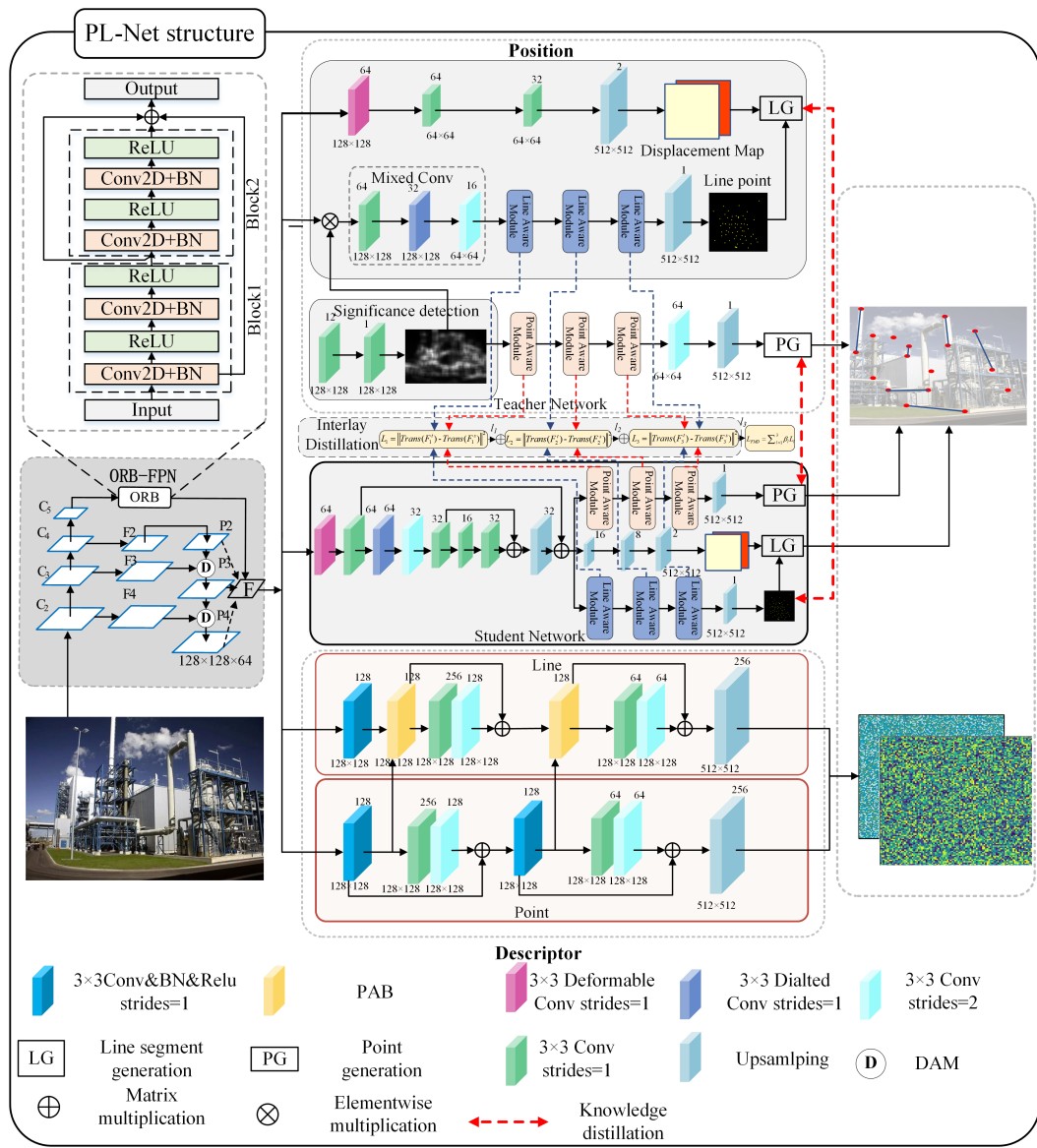

**Figure 2.** The framework of our proposed network PL-Net. We input two images of size $512 \times 512$ and then obtain key points, line segment, and their descriptors through the ORB-FPN, extraction module, and descriptor module. The gray box in the upper-right corner shows the structure of the ORB-FPN. The upper gray branch is responsible for extracting points and lines. The lower red branch is used to extract the point–line descriptor

In the neural network propagation process, the transmission of the signal from the first layer to the last layer is expressed as:

$$L_{i+1} = \sigma(U_i L_i) \tag{3}$$

where $L_{i+1}$ is the features of the $i+1$th layer in the network, and $\sigma$ represents the activation function. Suppose $U$ is a symmetrical positive definite matrix; let $V = \sqrt{U}$ and $\mu = VL$; then, for the nonlinear activation function $\sigma(\mu)$, there is a function $g(\mu)$, when $g'(\mu) = \sigma(\mu)$. We have:

$$\nabla \sum_i g\left(V_j^T \mu\right) = U\sigma\left(U^T L\right) = U\sigma(UL) \tag{4}$$

The objective function $f(\mu)$ is defined as:

$$f(\mu) = \frac{\|\mu^2\|}{2} - \sum_i g\left(V_j^T \mu\right) \tag{5}$$

where $V_i$ is the $i$th column of $V$. Then,

$$\nabla f(\mu_i) = \beta_i - V\sigma(V\mu_i) \tag{6}$$

Equation (2) can be expressed by:

$$\mu_{i+1} = \mu_i + \beta(\mu_i - \mu_{i-1}) - \alpha((1+\beta)\nabla f(\mu_i) - \beta\nabla f(\mu_{i-1})) \tag{7}$$

Recovering $L$ by $L = V^{-1}\mu$ leads to:

$$L_{i+1} = ((1+\beta)(1-\alpha) - \alpha\beta)L_i + \beta(1-\alpha)L_{i-1} + \alpha(1+\beta)\sigma(UL_i) \tag{8}$$

where $\sigma(UL_i)$ is the $i$th layer feed-forward network, and the ORB module structure is shown in Figure 2.

In order to aggregate the FPN multiscale feature information, a dual attention module (DAM) was designed to perform the feature aggregation. As shown in Figure 3, firstly, in order to obtain the position and channel information of the feature, the shallow feature map $x \in \mathbb{R}^{W \times H \times D_1}$ was passed through a global pooling operation and compression to generate the position vector $x_p \in \mathbb{R}^{W \times H \times 1}$ and the channel vector $x_c \in \mathbb{R}^{1 \times 1 \times D_1}$, respectively. Then, the position vector $x_p$ computed the weight of each position with a sigmoid activation function and multiplied it with the feature map $x_p$ to generate the spatial position feature map $F^p \in \mathbb{R}^{W \times H \times D_1}$, which was defined as:

$$F^p = \sigma(x) \otimes x \tag{9}$$

where $x$ is the shallow feature map, and $\sigma$ is the sigmoid activation function. Similarly, the convolution of the ReLU activation function, defined as $f(x) = max(0, x)$, and the sigmoid activation function was performed on the channel vector $x_c$. The weight of each channel was calculated and multiplied by the feature map $x$ to generate the feature map $F^c \in \mathbb{R}^{W \times H \times D_1}$, which was defined as follows:

$$F^c = \sigma(\delta(W_2(W_1(gp(x))))) \otimes x \tag{10}$$

where $\delta$ is the ReLU activation function, and $W_1$ and $W_2$ are convolution operations with sizes $1 \times 1 \times D_1/16$ and $1 \times 1 \times D_1$, respectively. Finally, the final output feature map $F$ was obtained by fusing the feature maps $F^p$, $F^c$, and $x'$. Then,

$$F = [(x \oplus F^p \oplus F^c), x'] \tag{11}$$

where $\oplus$ represents the addition of the corresponding elements of two matrices.

The input image was passed through the ORB-FPN module, which denoted the output of the backbone as $\{C_2, C_3, C_4, C_5\}$ with strides of $\{4, 8, 16, 32\}$. $\{F_2, F_3, F_4\}$ were obtained after a $1 \times 1$ convolution with the same 128-dimensional channel features. Finally, the ORB module was added to enhance the acceptance domain of the output feature by using the backbone network on $C_5$ to separate the important context information. After interpolation and maximum pooling of the extracted context features and the generated three feature maps, an elementwise summation was performed to obtain features $F$.

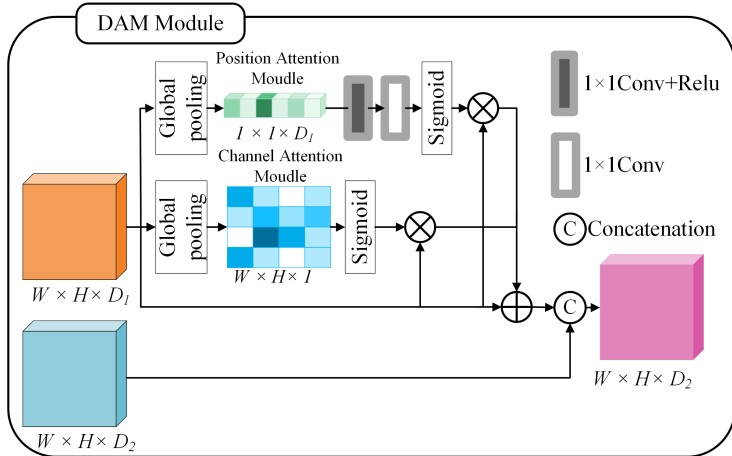

**Figure 3.** The dual attention module. The attention mechanism is adopted to adaptively aggregate different features, where the weights are normalized with the softmax function.

### 3.1.2. Key-point Detection Module

As shown in Figure 2, the key-point detection module consisted of three point perception modules and two $3 \times 3$ convolutions with a stride length of 1. A batch normalization layer and ReLU layers were added between each convolutional layer. The output vector was processed through a sigmoid activation function, so that the pixel values of the saliency map were between 0 and 1. Then, through the key-point perception module and convolution processing, the convolution operation was used to discriminate whether the $8 \times 8$ area prediction contained key points. Finally, the key points were detected by using the nonmaximum suppression (NMS) method in the key-point generation module (PG).

The point-aware module was used to capture the relationship between key points. As shown in Figure 4, the key-point extraction branch embedded a context enhancement module, which improved the feature expression ability. The output feature $y^p \in \mathbb{R}^{W \times H \times D}$ was obtained by fusing the convolutional features of different scales, which took $y^p \in {}^{W \times H \times D}$ as input. The above process was defined as:

$$y^p = W_1[x^p, BN(W_1 x^p), BN(W_2 x^p), BN(W_3 W_1 x^p)] \tag{12}$$

where $x^P$ is the input feature, $W_1$, $W_2$, and $W_3$ are convolution operations with sizes of $1 \times 1$, $2 \times 2$, and $3 \times 3$, respectively. *BN* is a normalization, and $[]$ is a splicing operation.

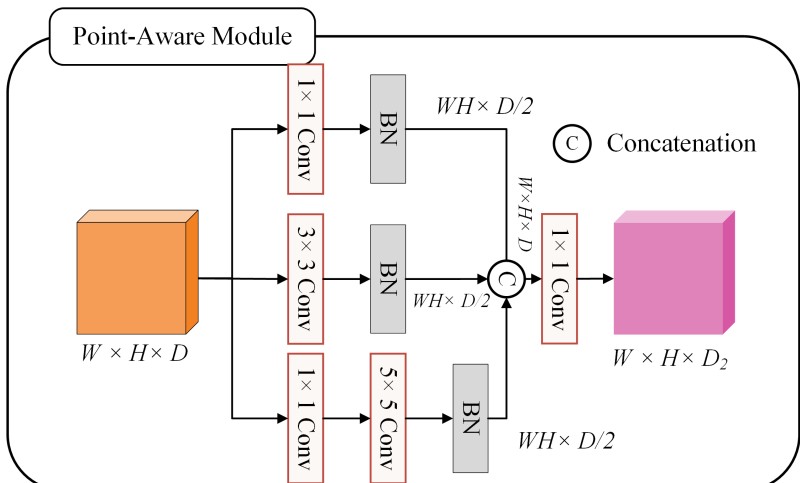

**Figure 4.** The point-aware module. BN means batch normalization.

### 3.1.3. Line-Segment Detection Module

The line-segment detection module extracted the features of the plane image through the ORB-FPN module and then input it to the line-segment extraction module to generate a midpoint with two symmetrical endpoints as the line-segment detection result. The extraction of the line segment's center point [28] used the classification model to judge whether the pixel was the center point of the line segment. Since the shape of the line segment was narrow and long, a large receptive field was required to classify the center of the line segment. Therefore, a hybrid convolution module was introduced by stacking three convolutional layers, a $3 \times 3$ deformable convolutional layer and two $3 \times 3$ dilated convolutional layers. The receptive field of the network was increased while reducing the parameters of the network. Then, three line-segment perception modules were used to enhance the feature representation ability. Finally, a deconvolution layer was used to restore the size of the output map to $512 \times 512$, which represented the center point of the line segment on the output feature map. The displacement regression task of the line-segment extraction branch was designed to predict the angle and length of the end point relative to the midpoint. It was composed of a $3 \times 3$ deformable convolution layer and two $3 \times 3$ convolutional layers with a stride of 1. The relevant displacement was indexed by the position through the output map. Finally, the CAL [29] was used for the line-segment generation, and the two endpoints of the line segment were defined as follows:

$$(x_{l_s}, y_{l_s}) = (x_{l_c}, y_{l_c}) + \frac{\alpha}{2}(\cos\theta, \sin\theta) \tag{13}$$

$$(x_{le}, y_{l_e}) = (x_{l_c}, y_{l_c}) - \frac{\alpha}{2}(\cos\theta, \sin\theta) \tag{14}$$

where $(x_{l_c}, y_{l_c})$ is the coordinates of the root node, $\alpha$ is the length of the line segment, and $\theta$ is the rotation angle.

In the line-segment detection, a line-aware module was introduced to effectively extract line shape features. As shown in Figure 5, the line-segment-aware module adopted the improved self-attention mechanism, which took the feature $x^L \in \mathbb{R}^{W \times H \times D}$ as input and fused the self-attention features to generate the final output feature $y^L \in \mathbb{R}^{W \times H \times D}$; the above process was defined as follows:

$$y^L = x^L \oplus W_1(\alpha(W_q x^L \times W_k x^L) \times W_v x^L) \tag{15}$$

where $x^L$ is the input feature, $W_1$, $W_q$, $W_k$, and $W_v$ are the learned weight matrices, which were implemented as $1 \times 1$ convolutions, and $\oplus$ means that the corresponding elements of the two matrices are added.

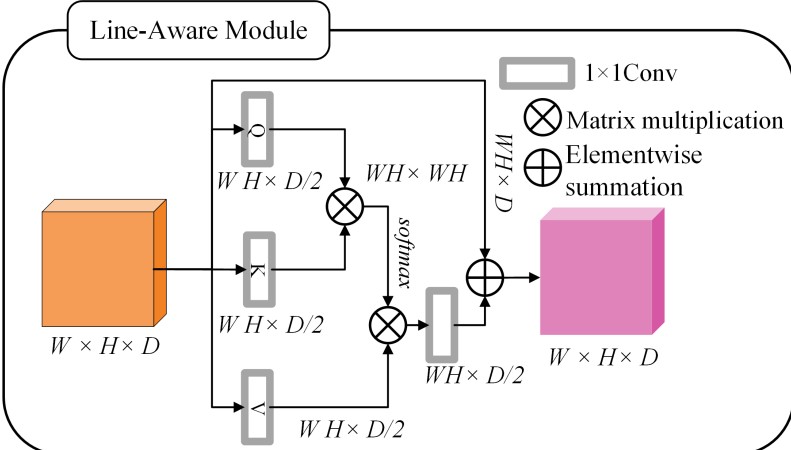

**Figure 5.** The line-aware module.

### 3.1.4. Parallel Attention Module

To efficiently merge the two branches of point descriptors and line descriptors, we designed a parallel attention module (PAB) based on self-attention and channel attention. As shown in Figure 6, the output features of the key-point description branch contained line-edge information with a strong correlation. In order to improve the accuracy of the line descriptor, a lightweight attention mechanism was used to assign more weights for the features of useful regions. The output feature map of the point description branch was expressed as $X_E \in \mathbb{R}^{C \times H \times W}$. A one-dimensional convolution of $X_E$ was performed to obtain the spatial attention map $A_E \in \mathbb{R}^{C \times H \times W}$. The edge feature map $X_E^S \in \mathbb{R}^{C \times H \times W}$ was calculated as: $X_E^S = a(X_E \odot A_E) + X_E$, where $a$ is a learnable parameter that was initialized to 0. The CAEU module was designed to calculate a channel attention map, which recalibrated the weight of the channel and obtained the fused feature map $X_F^S \in \mathbb{R}^{C \times H \times W}$ from $X_F^S = X_E^S \otimes \delta(Conv1 \times 1(Conv1 \times 1(GAP(X_E^S))))$. Finally, the final output $X_F^{SC} \in \mathbb{R}^{2C \times H \times W}$ of PAB was obtained by concatenating $X_T^S$ and $X_F^S$ together.

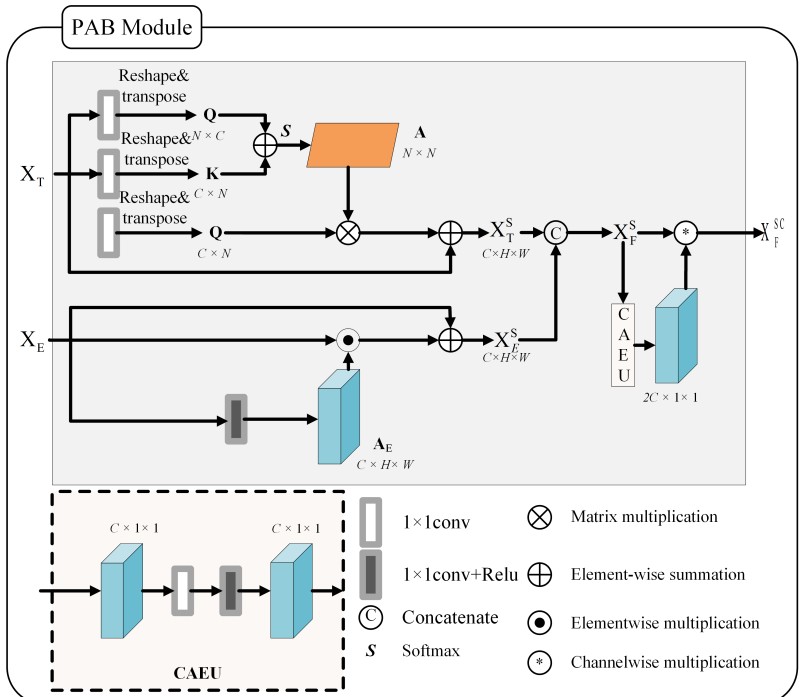

**Figure 6.** The parallel attention block (PAB). The PAB is designed to transfer important information to the line branch output ($X_F^{SC}$). The pink box in the lower-left corner shows the structure of the CAEU.

### 3.1.5. Network Output Distillation

In order to reduce the increasing computation cost caused by the introduction of the attention mechanism and the point–line perception module, we further compressed the point–line detection model PL-Net. A transfer-aware method is presented to transfer the information from the teacher model to the student network. The knowledge distillation strategy (KD) [30–32] was used to fine-tune the accuracy of the recovery model. As shown in Figure 2, the multiples tasks were combined with the teacher network and the student network to guide the training of the student feature extraction. In the training process, the adaptive weighted multitask distillation was realized, and $X_{tea}$, $Y_{tea}$, and $Z_{tea}$ represented the key point of the teacher model, the center point of the line segment, and the output of the line-segment regression feature layer, respectively. The student model corresponded to

the feature layers X$_{stu}$, Y$_{stu}$, and Z$_{stu}$. The mean squared error (MSE) function was applied to the multitask distillation. The loss function of the training distillation was:

$$\begin{cases} L_p^S = \|X_{tea} - X_{stu}\|^2 \\ L_{root}^S = \|Y_{tea} - Y_{stu}\|^2 \\ L_{dis}^S = \|Y_{tea} - Y_{stu}\|^2 \end{cases} \tag{16}$$

where $L_p^S$, $L_{root}^S$, and $L_{dis}^S$ denote the key point, the line segment's center point, and the distillation loss of the line-segment regression task, respectively. Then, the weighted distillation loss was defined as:

$$L_{MSE}^s = \sum_l \omega_l L_l^s \tag{17}$$

where $L_{MSE}^s$ is the multitask distillation loss, and $\omega_l$ represents the weight value of the verification loss.

### 3.1.6. Interlayer Knowledge Distillation

Different from the network output distillation, the student network was further enhanced by performing an interlayer knowledge distillation between the output of the teacher model and the student model. The aware modules of each student layer were associated with the relevant target-layer-aware modules for knowledge transfer. The layer's knowledge distillation loss was defined as:

$$L_{FMD} = \sum_{(s_l, t_l) \in C} Dist(Trans^t(F_{tl}^t), Trans^s(F_{sl}^s)) \tag{18}$$

Then, the overall loss was obtained as:

$$L_{total} = L_{MSE}^s + \beta L_{FMD} \tag{19}$$

where $Trans(\cdot)$ means to convert the feature map of the perception module into a specific manual representation through the attention map. $C$ is the perception module, and $F_{tl}^t$ and $F_{sl}^s$ are the feature layers of the $l$th layer of the student model and the teacher model, respectively. The distance function $Dist(\cdot)$ was used for computing the distillation loss of the feature maps, and $\beta$ was a hyperparameter.

### 3.2. Heterogeneous Graph Attention Network

As shown in Figure 7, consider two images $I$ and $I'$, and the number of two feature sets $m$ and $n$ belonging to them, respectively. Let $d \in \mathbb{R}^D$ be the feature descriptor, where $D$ is the dimension of the descriptor. We utilized an attention graph neural network to integrate the contextual cues and enhance its feature expression ability.

Position encoders were used for the two input features, and the key points and lines positions were embedded into a high-dimensional vector by adding position encoding to $\widehat{F}_1$ and $\widehat{F}_2$; thus, we had:

$$f_i = d_i + \text{MLP}(P_i) \tag{20}$$

$$\text{MLP}(P_i) = W_a \sigma(W_b P_i) \tag{21}$$

where $P_i$ is the position of the feature, $d_i$ is the descriptor information, and $\sigma(\cdot)$ is the ReLU activation function.

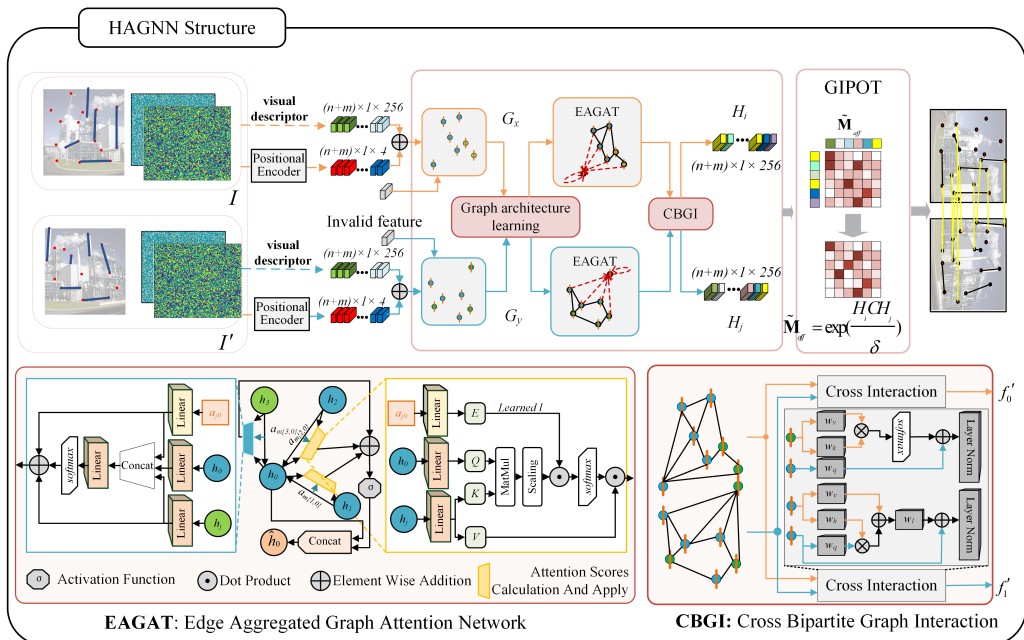

**Figure 7.** The overall structure of HAGNN. The first stage embeds the key points and line positions $I$ and $I'$ into a high-dimensional vector, which generates $G_x$ and $G_y$. Graph architecture learning is used to construct the graph. The two graphs can generate discriminative features through EAGAT and CBGI. The second stage computes the affinity matrix and the assignment matrix between two sets $H_i$ and $H_j$ and uses the assignment matrix to find matches and filter nonmatches.

### 3.2.1. Edge-Clustering Graph Attention Module

We propose an edge-aggregated graph attention network (EAGAT) based on GAT [33], which uses edge information for feature enhancement during the aggregation process. In order to make full use of the information of edge features, these different types of links used different attention mechanisms. For features of the same nature (points and points, lines and lines), the self-attention mechanism was used for the aggregation. For features of different nature (points and lines), the cascade method was used for the aggregation. Let the feature of the vertex $v'_i$ in the graph be $f_i$, defined as:

$$
\begin{aligned}
f'_i = \sigma\big( &\sum_{j \in \mathcal{N}_s} softmax(\frac{W^a f_i W^\beta f_j}{\sqrt{d_k}} W^\gamma a_{ji}) W^\varepsilon f_j \\
+ &\sum_{j \in \mathcal{N}_d} softmax([W^a f_i || W^\beta f_j || W^\gamma a_{ji}]) W^\varepsilon) || f_i
\end{aligned}
\tag{22}
$$

where $W^a$, $W^\beta$, $W^\gamma$, and $W^\varepsilon \in \mathbb{R}^{f'_\eta \times f_\eta}$ represent the weight parameters, $\mathcal{N}_s$ is the feature set of the same nature, $\mathcal{N}_d$ represents the feature set of different properties, and $\sigma$ is the ReLU activation function.

### 3.2.2. Cross-Heterogeneous Graph Iteration Module

Due to the affinity learning problem of message passing between graphs in graph matching, a point–line heterogeneous graph message-passing method is proposed to enhance the node features through an interactive correlation. The edge features and node features were aggregated in two ways. For nodes of the same nature (points and points, lines and lines), we used linear attention for the aggregation, and for nodes of

different properties (points and lines), we used the aggregation method of the self-attention mechanism. Then, $f'_{si} \in \mathbb{R}^{f'\eta}$ is expressed as:

$$f'_{si} = \mathrm{LN}\{ \sum_{j \in \mathcal{N}_{di}} (softmax[(W_v f_j)(W_k f_j)^T] + W_q f_{si})\}$$
$$+ \mathrm{LN}\{ \sum_{j \in \mathcal{N}_{si}} (W_l((W_q f_{si})(W_k f_j)^T + W_k f_j) + f_{si}\} \tag{23}$$

where $v_{l/r} \in \mathbb{R}^{f'\eta}$ is the feature node of the two graphs, $W_{(\cdot)}$ is the weight parameter, and LN represents a layer normalization (LN).

*3.3. Greedy near Iterative Matching Module*

The output $H_i$ of the last layer in the graph neural network is the feature of graph $I'$. $H_j$ is the feature of graph $I'$, and the point–line distance matrix $G \in \mathbb{R}^{+N_1 \times N_2}$ can be expressed as:

$$G = f_{aff}(H_i, H'_j), i \in v_1, j \in v_2 \tag{24}$$

where $f_{aff}$ is the weighted bilinear function, defined as:

$$f_{aff}(H_i, H'_j) = \exp(\frac{H_i^T K H_j^T}{\tau}) \tag{25}$$

where the feature is an *n*-dimensional vector, namely $\forall i \in v_1, j \in v_2$ and $H_i^T, H_j^T \in \mathbb{R}^{n \times n}$, $K \in \mathbb{R}^{n \times n}$ is the weight matrix of the affinity function, and $\tau$ is the regularization parameter. In the matching process, due to the inconsistency of point and line types, a direct fusion may cause a mismatch of point and line types. For this reason, we regarded the unit block diagonal matrix as the initial coupling matrix $\Gamma^{(1)}$ so that the relationship between point features and line features in the iterative process minimized the matching cost. Then, we had:

$$\Gamma^{(1)} \leftarrow \begin{bmatrix} 11_p & 0 \\ 0 & 11_L \end{bmatrix} \tag{26}$$

where $P$ is the number of points after completion, and $L$ is the number of lines after completion. Then, the point–line discrete distribution Sinkhorn distance [34,35] was defined as:

$$W_{\in}(u, v) = \min_{\Gamma \in \sum(u,v)} \langle C, \Gamma \rangle + \lambda h(\Gamma) \tag{27}$$

where *u* and *v* are probability vectors, $W_{\in}(u, v)$ is the distance between *u* and *v*, the matrix $C = [c_{ij}] \in \mathbb{R}^{+n \times n}$ is the cost matrix, and $c_{ij}$ is the distance between $u_i$ and $v_j$. The regularization term $h(\Gamma) = \sum_{i,j} \Gamma_{ij} \ln \Gamma_{ij}$. The proximal point iteration method [35] was used to solve Equation (25). According to the proximal point iteration method, it is defined as the Bregman divergence:

$$D_h(x, y) = \sum_{i=1}^n x_i \ln \frac{x_i}{y_i} - \sum_{i=1}^n x_i - \sum_{i=1}^n y_i \tag{28}$$

After introducing the near-end point iteration, Equation (27) can be rewritten as:

$$\Gamma^{(t+1)} = \underset{\Gamma \in \sum(u,v)}{\mathrm{argmin}} \langle C, \Gamma \rangle + \beta^t D_h(\Gamma, \Gamma^{(t)}) \tag{29}$$

Substituting the Bregman divergence Equation (28) into Equation (29), it becomes

$$\Gamma^{(t+1)} = \underset{\Gamma \in \sum (u,v)}{\operatorname{argmin}} \langle C', \Gamma \rangle + \beta^t h(\Gamma) \tag{30}$$

where $C' = C - \beta^t \ln \Gamma^{(t)}$; we used the greedy strategy to update the best row or column and defined the distance matrix

$$\rho(x, y) = y - x + \log \frac{x}{y} \tag{31}$$

According to Equations (29) and (30), the affinity matrix was updated to find the best matching relationship, and the specific algorithm flow is shown in Algorithm 1.

---

**Algorithm 1:** GIPOT($\mu$, $v$, $G$).

**Input:** Point–line features of graph network output $H_i$ and $H_j$

**Output:** $\Gamma^{(t+1)}$

initialize $\Gamma^{(1)} \leftarrow \begin{bmatrix} 11_p & 0 \\ 0 & 11_L \end{bmatrix}$, $G \leftarrow f_{aff}(H_i, H'_j)$

**begin**

    **for** $t = 1, 2, 3 \ldots$ **do**

        $Q \leftarrow G \odot \Gamma^{(t)}$

        $I \leftarrow \operatorname{argmax}_i \rho(u_i, u_i(Q))$

        $J \leftarrow \operatorname{argmax}_j \rho(u_j, u_j(Q))$

        $\Gamma^{(1)} \leftarrow diag(\exp(a))Qdiag(\exp(b))$

        **if** $\rho(u_I, u_I(Q)) > \rho(u_J, u_J(Q))$ **then**

            $a_I \leftarrow a_I + \log \frac{u_I}{u_I(Q)}$

        **else**

            $a_J \leftarrow a_J + \log \frac{u_J}{u_J(Q)}$

        **end**

        $\Gamma^{(1)} \leftarrow diag(\exp(a))Qdiag(\exp(b))$

    **end**

**end**

---

### 3.4. Loss Function

In order to realize the matching of points and lines, the point–line extraction loss, descriptor extraction loss, and point–line matching loss were used as the loss functions during model training.

#### 3.4.1. Point–Line Extraction Loss

In the training stage of the point and line extraction branch, the output included the root node's confidence map, key-point map, and displacement map. The losses of these three tasks were combined into Equation (32), defined as follows:

$$L_{PLE} = L_{root} + L_p + L_{dis} \tag{32}$$

The ground truth of the root-point confidence map was constructed by marking the root-point positions on a zero map. A weighted binary cross-entropy loss $L_{root}$ was used to supervise this task, which was defined as:

$$L_{root} = -\sum_i \widetilde{R}_i \log R_i + (1 - \widetilde{R}_i) \log(1 - R_i) \tag{33}$$

where $\widetilde{R}_i$ and $R_i$ are the prediction and true label of the root node of the line segment, respectively. The true value of the key point was marked with the ORB feature. $L_p(X,Y)$ was defined as:

$$L_p(X,Y) = \frac{1}{H_c w_c} \sum l_p\left(T^{ij}, \widetilde{T^{ij}}\right) \tag{34}$$

$$l_p\left(T^{ij}, \widetilde{T^{ij}}\right) = -\log\left(\frac{\exp(T^{ij}_k)}{\sum_1^{64} \exp(T^{ij}_k)}\right) \tag{35}$$

The displacement part of the line segment relative to the root node was used to locate the length and angle of the line segment, which used the L1 loss and L1 smoothing loss, respectively defined as:

$$L_{dis} = \sum_{i=1}^{m} \begin{cases} \left|\theta^i - \hat{\theta}^i\right| + 0.5 * (\rho^i - \hat{\rho}^i)^2 & if \left|\rho^i - \hat{\rho}^i\right| < 1 \\ \left|\theta^i - \hat{\theta}^i\right| + \left|\rho^i - \hat{\rho}^i\right| - 0.5 & otherwise \end{cases} \tag{36}$$

where $\theta^i$ and $\rho^i$ represent the actual line segment's length and angle, and $\hat{\theta}^i$ and $\hat{\rho}^i$ are the predicted line segment's length and angle, respectively.

### 3.4.2. Point–Line Descriptor Loss

Denote the original image as $I$, and apply the homography transformation for $I$ to form a new image $I'$. Since the homography transformation is known, the corresponding relationship between key points and line segments on $I$ and $I'$ can be obtained. Therefore, the loss function can be defined as:

$$L_d\left(\theta, \left\{d_a^\theta, d_+^\theta, d_-^\theta\right\}\right) = [m + \left\|d_a^\theta - d_+^\theta\right\|^2 - \min_{d_-^\theta \in d_-^\theta} \left\|d_a^\theta - d_-^\theta\right\|^2]_+ \tag{37}$$

where the parameter $m$ was set to 0.5, $d_a^\theta$ is the descriptor on $I$ of the anchor point, $d_+^\theta$ is the matching descriptor on $I'$ of the positive sample, and $d_-^\theta$ is the set of nonmatching descriptors on $I'$ of the negative sample.

### 3.4.3. Matching Loss

For a matching network using the L2 loss, the loss function can be expressed as:

$$L_m = \frac{1}{|M_{gt}|} \sum_{(i,j) \in M_{gt}} \frac{1}{\sigma^2(i)} \left\|M(i,j) - M_{gt}(i,j)\right\|_2 \tag{38}$$

where $\sigma^2(i)$ is the confidence variance of feature $i$. $M(i,j)$ is the matching probability of feature $i$ and feature $j$, and $M_{gt}$ is the real-valued matrix obtained by the homography transformation.

### 3.4.4. Normalization

The total loss function was the sum of the above loss functions:

$$L_{sum} = \lambda_1 L_{PLE} + \lambda_2 L_d + \lambda_3 L_m \tag{39}$$

where $\lambda_1$, $\lambda_2$ and $\lambda_3$ represent the coefficients of each loss function, respectively. $\lambda_{1,2,3} = \{0.25, 0.25, 0.5\}$.

## 4. Experiments and Evaluation
### 4.1. Model Training Details

A wide range of experiments were performed on different datasets to demonstrate the efficacy of our method. Our approach was evaluated with several evaluation criteria by comparing to the typical SLAM methods on the KITTI dataset [36]. We used the training set of Wireframe [37] with the ground truth to train our models and the other compared

methods. The training process used data enhancement techniques such as random Gaussian noise, motion blur, and brightness level changes to improve the network's robustness ability for changes in lighting and viewing angles. For end-to-end training of the point–line matching network, our network was implemented in Pytorch [38] using the Adam [39] optimizer to train the network with an initial learning rate of $1 \times 10^{-5}$ and a decay of the learning rate by 20 at each epoch. We trained our model on a GeForce GTX2080Ti GPU.

### 4.2. KITTI Dataset Evaluation

We tested the proposed algorithm on the KITTI dataset [36]. The quantitative evaluations for the different SLAM systems were the absolute trajectory error (ATE) [40] and the relative pose error (RPE) [41] based on translations and rotations. Table 1 shows the performance of this system was better than ORB-SLAM2, especially in sequences with strong lighting, motion blur, and low texture areas, such as 06 and 09. It can be seen that the multifeature fusion not only improved the accuracy of the algorithm but also avoided the degradation problem that may occur in the pose solution algorithm when using a single feature.

**Table 1.** Comparison between ATE and RPE on different SLAM algorithms.

| Seq | ORB-SLAM2 | | | Our | | |
|-----|-----------|--------------------|-------------------|---------|--------------------|-------------------|
|     | ATE (m)   | $RPE_{trans}$ (%)  | $RPE_{rot}$ (deg/m) | ATE (m) | $RPE_{trans}$ (%)  | $RPE_{rot}$ (deg/m) |
| 00  | 1.266     | 52.5               | 0.363             | 1.233   | 2.9                | 0.122             |
| 01  | 4.296     | 3.4                | 0.420             | 2.616   | 4.8                | 0.044             |
| 02  | 12.790    | 4.3                | 0.107             | 12.721  | 3.6                | 0.077             |
| 03  | 0.403     | 0.8                | 0.072             | 0.385   | 2.0                | 0.055             |
| 04  | 0.466     | 2.2                | 0.055             | 0.192   | 2.1                | 0.040             |
| 05  | 0.348     | 2.3                | 0.144             | 0.402   | 1.7                | 0.056             |
| 06  | 1.184     | 3.9                | 0.089             | 0.572   | 1.8                | 0.042             |
| 07  | 0.439     | 1.3                | 0.076             | 0.436   | 1.6                | 0.046             |
| 08  | 3.122     | 12.1               | 0.076             | 2.874   | 3.9                | 0.054             |
| 09  | 3.319     | 15.0               | 0.104             | 1.537   | 2.2                | 0.054             |
| 10  | 0.927     | 2.6                | 0.090             | 0.989   | 2.1                | 0.060             |

Figure 8 shows the comparison results of ORB-SLAM2 and the method in this paper on KITTI's partial sequences. It can be seen that the algorithm of this paper was equivalent to ORB-SLAM2 as a whole. However, in sequences such as 00, 02, and 09, it showed that our method had the best results. On the other hand, the ORB-SLAM2 lost track easily and did not have a whole trajectory.

Figure 9 depicts the variation curve of the pose estimation error with the number of training iterations. When we used more than 80 iterations, the rotation and translation errors were relatively small while the performance improved, which showed that the algorithm in this paper had a good convergence.

Figure 10 visualizes the statistical error property between our system and ORB-SLAM2 on sequence 09 of KITTI. It shows that our model achieved obviously a better performance than ORB-SLAM2 in terms of the root-mean-square error (controlled within 2 m), extreme value error, and the sum of the squared errors. Figure 10b shows the columnar statistical comparison of postural errors; our model was better than ORB-SLAM2 in the APE distribution range of the algorithm.

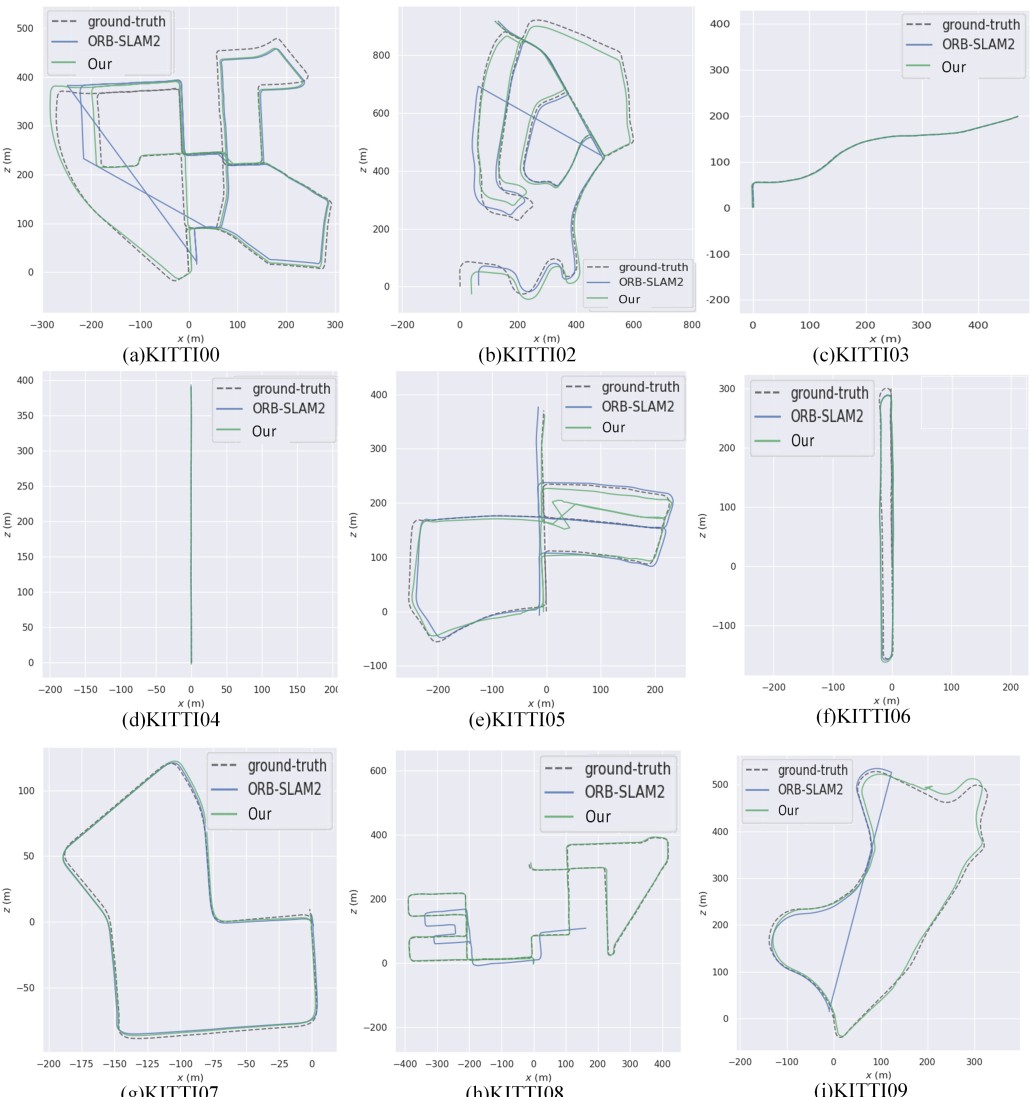

**Figure 8.** Comparison of trajectories estimated by our SLAM method, ORB-SLAM2, and the ground truth on the KITTI dataset.

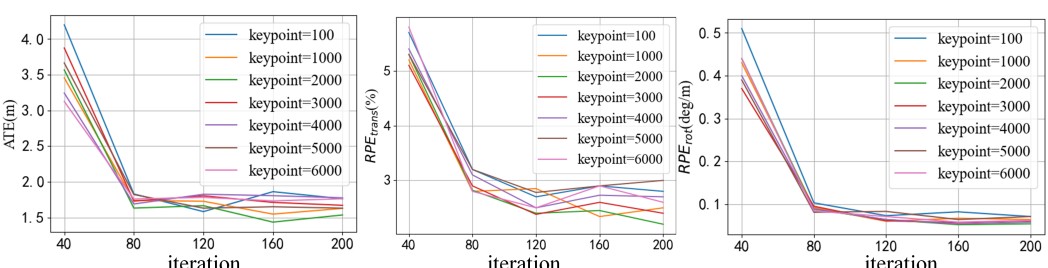

**Figure 9.** The variation curve of the pose estimation error with the number of training iterations on sequence 09 of KITTI. Each color represents a different number of sampling points.

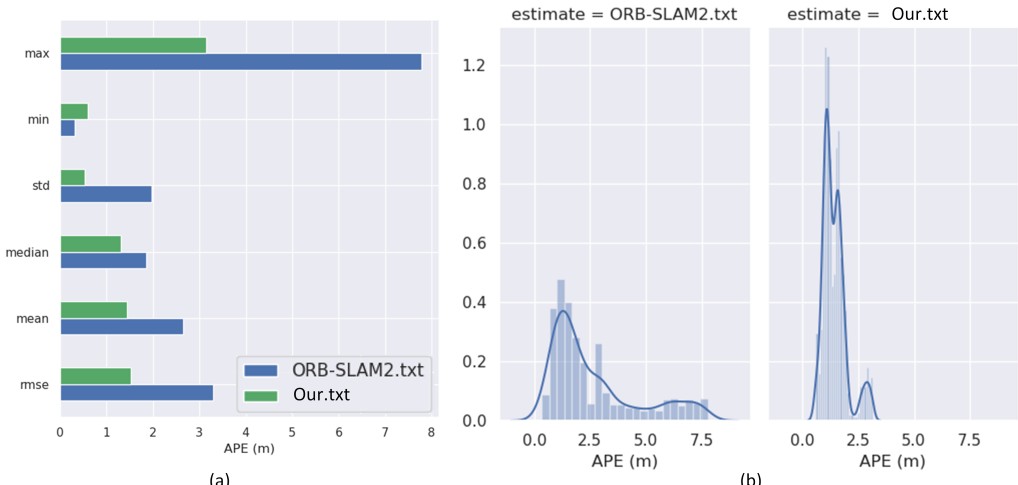

(a)  (b)

**Figure 10.** Comparison of the statistical error property between our system and ORB-SLAM2 on sequence 09 of KITTI. (**a**) Quantitative index chart of ATE. (**b**) Histogram of ATE.

### 4.3. Real Data Evaluation

As shown in Figure 11, the feasibility of the algorithm was verified on a physical and virtual oil and gas station, where the virtual simulation platform with a quadruped robot was built with Unity3D.

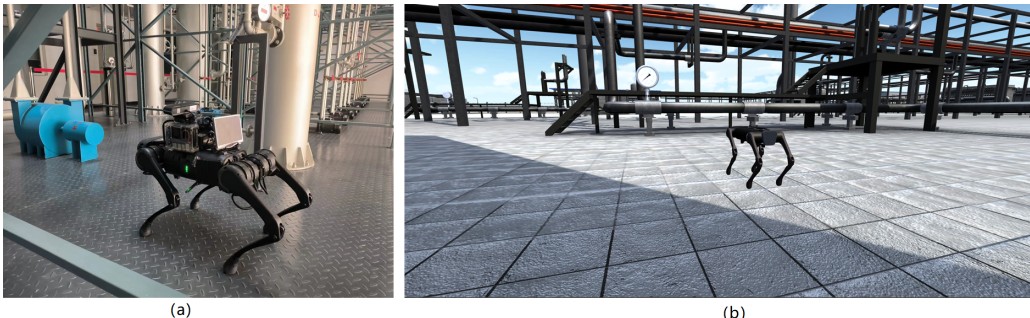

(a)  (b)

**Figure 11.** The experimental robot platform. (**a**) The physical experimental robot platform with robot hardware including a camera, depth camera, and IMU. An additional GPS/RTK was used for the ground-truth estimation. (**b**) The virtual simulation platform.

To show the effects of point and line features on the SLAM system, we intercepted two frames of images for extracting the features by PL-Net and matching by HAGNN. As shown in Figure 12, it can be seen that the combination of point and line could make the SLAM algorithm obtain richer and more diverse feature information.

The virtual simulation platform used a quadruped robot to inspect the oil field equipment. As shown in Figure 13, this trajectory was compared with the ground truth by using the evaluation package to obtain the RPE and APE. It can be seen that the area with a larger error was basically distributed at the corner of the trajectory. The RMSE was 17.5 m. Overall, the trajectory of our method was consistent with the ground truth with a high accuracy.

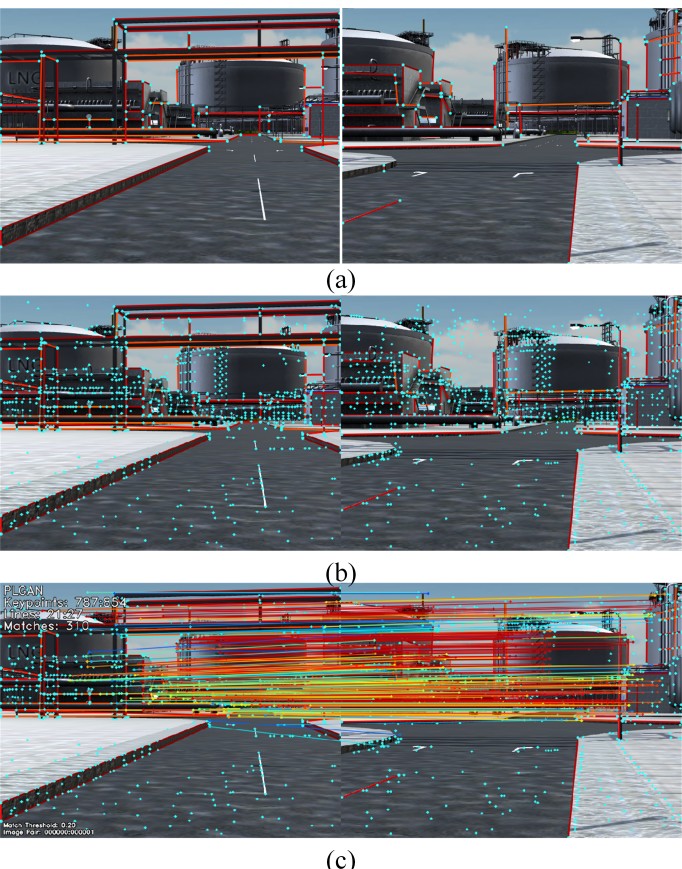

**Figure 12.** The effect of point–line feature tracking. (**a**) The line-segment extraction results. (**b**) The point and line-segment extraction results. (**c**) The point and line matching results. We visualized the matching results with RGB color.

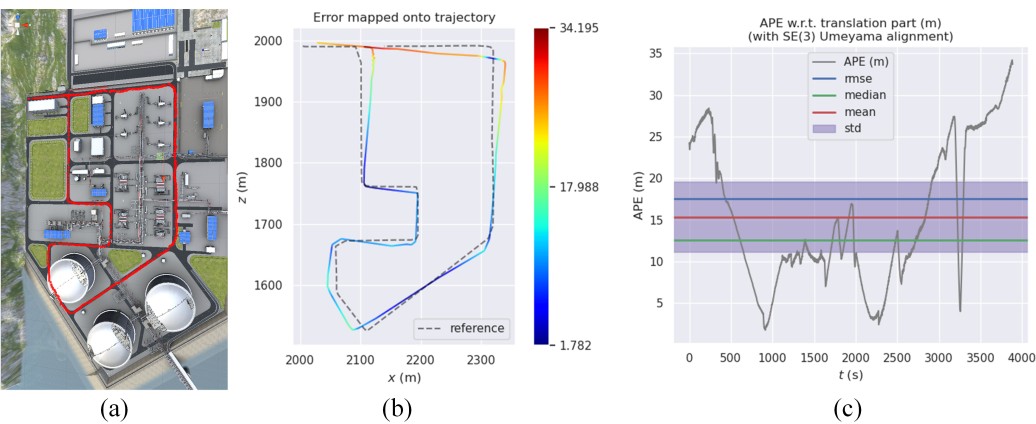

**Figure 13.** Simulation platform experiment. (**a**) The simulation platform. (**b**) The error mapped onto trajectory. (**c**) RMSE of ATE in meters after translation and scale alignment.

## 4.4. GIPOT Experiment

In order to illustrate the convergence of GIPOT with different $\beta$, the Wasserstein distance of two one-dimensional Gaussian distributions was measured as an evaluation index. As shown in Figure 14a, the blue equation was $0.5N(70, 8) + 0.5(35, 10)$, the red equation was $0.4N(80, 9) + 0.6N(40, 10)$, where $N(\mu, \sigma^2)$ is the probability density function of the one-dimensional Gaussian distribution, $\mu$ and $\sigma^2$ are the mean and variance, respectively. Figure 14b shows the convergence of GIPOT under different conditions. Compared with the Sinkhorn method, the convergence of the GIPOT iteration was quicker when $\beta$ was large.

GIPOT could converge to the exact Wasserstein distance with a complexity comparable to that of Sinkhorn.

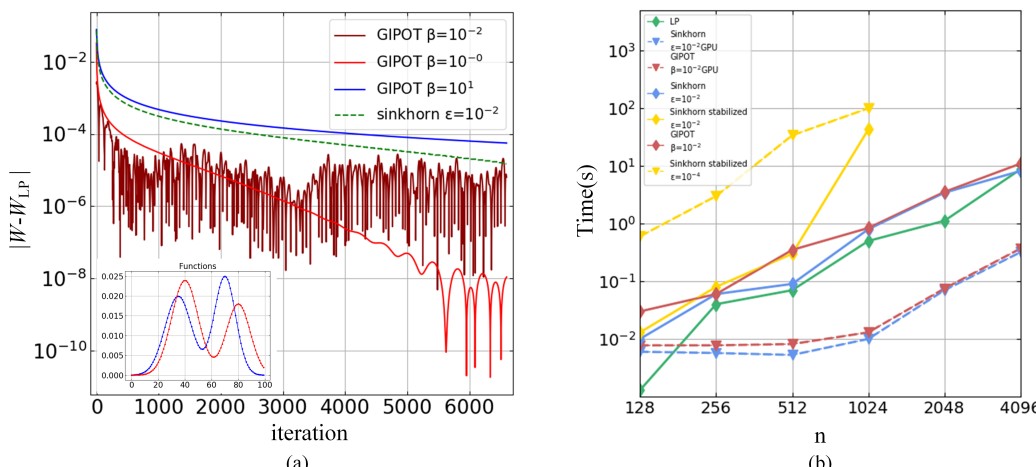

**Figure 14.** The difference graph of the Wasserstein distance. (**a**) GIPOT under different conditions and convergence trajectory graph. We also plotted the ones for the Sinkhorn method for comparison. (**b**) The average time of GIPOT and Sinkhorn iterations under different conditions.

### 4.5. Ablation Study

We used two datasets to demonstrate our proposed knowledge distillation. The result of the teacher network and the student network are shown in Figure 15. Compared with the true value, both models could identify key points and line segments with high precision. Although there were some small missing line segments and connection errors in the results of the student network, the expression of the line-segment structure in the environment was basically accurate. The quantitative comparison is shown in Table 2. Although the performance of the student network was slightly lower than that of the teacher network, the operation speed was increased by 73%.

**Table 2.** Quantitative evaluation of PL-Net point-line detection knowledge distillation method on Wireframe dataset and YorkUrban dataset.

| Method | Wireframe dataset | | | YorkUrban dataset | | | FPS |
|---|---|---|---|---|---|---|---|
| | $F^H$ | $sAP$ | $LAP$ | $F^H$ | $sAP$ | $LAP$ | |
| Student | 77.5 | 58.9 | 59.8 | 64.6 | 25.9 | 32.0 | 12.5 |
| Teacher | 80.6 | 57.6 | 61.3 | 67.2 | 27.6 | 34.3 | 7.2 |

To verify the role of the multifeature fusion in the SLAM system, the root-mean-square error (RMSE) of the ATE index under different feature combinations was calculated. The experimental results are given in Table 3, where the point–line feature combination method used in this paper significantly improved the accuracy of the pose estimation. To evaluate our design decisions, we evaluated four different variants with results. This ablation study, presented in Table 4, showed that all HAGNN blocks were useful and brought substantial performance gains. "No EAGAT" replaced all EAGAT layers with CHGI layers, and the matching accuracy decreased by 9.7%. "No CHGI" replaced all CHGI layers with EAGAT layer, and the precision of the resulting matching decreased by 22.6%, "No HAGNN" replaced the graph neural network with a single linear projection, and the precision of the resulting matching decreased by 26.1%.

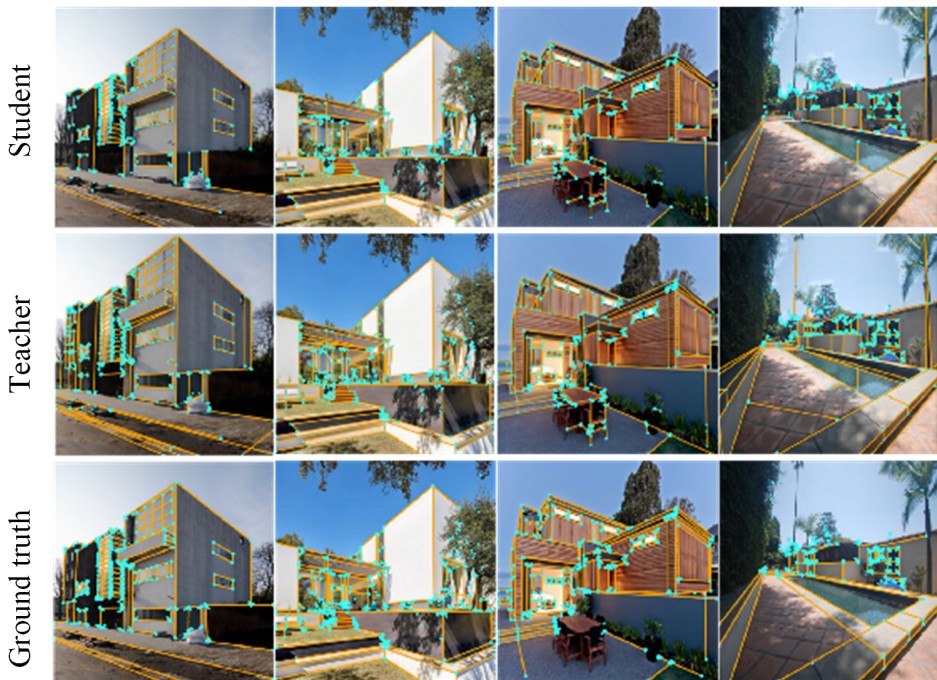

**Figure 15.** Qualitative evaluation of PL-Net point–line detection knowledge distillation method on the Wireframe dataset and the YorkUrban dataset.

**Table 3.** Results of ablation experiment in term of the RMSE of ATE (Unit: m).

| Seq | P-SLAM | L-SLAM | PL-SLAM | ORB-SLAM2 | LSD-SLAM | PTAM |
| --- | --- | --- | --- | --- | --- | --- |
| 00 | 1.203 | 6.233 | 1.233 | 1.266 | 5.347 | 2.842 |
| 01 | 3.934 | 12.367 | 2.616 | 4.296 | — | 3.358 |
| 02 | 7.689 | — | 12.721 | 12.790 | — | 13.742 |
| 03 | 0.393 | 5.457 | 0.385 | 0.403 | 7.431 | 2.302 |
| 04 | 0.347 | 13.824 | 0.192 | 0.466 | — | 2.773 |
| 05 | 0.863 | — | 0.402 | 0.348 | 1.293 | 0.456 |
| 06 | 0.884 | — | 0.572 | 1.184 | — | 1.024 |
| 07 | 0.255 | — | 0.436 | 0.439 | — | 0.423 |
| 08 | 3.122 | — | 2.874 | 3.122 | — | 3.358 |
| 09 | 2.625 | 4.783 | 1.537 | 3.319 | 11.395 | 2.048 |
| 10 | 0.447 | 5.824 | 0.989 | 0.927 | 2.841 | 0.768 |

The proposed HAGNN was compared with two feature matching methods: the nearest neighbor (NN) method and SuperGlue. As show in Table 5, it can be seen clearly that HAGNN had a significantly higher pose estimation accuracy than all competitors, which showed a higher feature expression ability.

**Table 4.** Ablation of HAGNN.

| | Known | | Unknown | |
| --- | --- | --- | --- | --- |
| | Match Precision | Matching Score | Match Precision | Matching Score |
| No EAGAT | 79.6 | 29.5 | 55.3 | 15.6 |
| No CHGI | 66.7 | 25.3 | 48.2 | 18.5 |
| No HAGNN | 63.2 | 19.4 | 51.2 | 10.3 |
| Full | 89.3 | 34.2 | 78.3 | 23.8 |

**Table 5.** Experimental results of the pose estimation. Matching PL-Net features with HAGNN resulted in a significantly higher pose accuracy (AUC), precision (P), and matching score (MS) than with handcrafted or other learned methods.

| Feature | Matcher | Pose Estimation AUC | | | P | MS |
|---|---|---|---|---|---|---|
| | | @5° | @10° | @20° | | |
| SIFT | NN | 7.89 | 10.22 | 35.30 | 43.4 | 1.7 |
| SIFT | SuperGlue | 23.68 | 36.44 | 49.44 | 74.1 | 7.2 |
| SuperPoint | NN | 9.80 | 18.99 | 30.88 | 22.5 | 4.9 |
| SuperPoint | SuperGlue | 34.18 | 44.32 | 64.16 | 84.9 | 11.1 |
| LSD + LBD | NN | 5.43 | 7.83 | 28.54 | 32.5 | 1.3 |
| SOLD$^2$ | NN | 18.34 | 13.22 | 23.51 | 63.6 | 6.2 |
| SuperPoint + SOLD$^2$ | Ours | 35.86 | 44.73 | 64.43 | 85.3 | 12.3 |
| Ours | Ours | 36.67 | 44.26 | 64.73 | 86.6 | 12.7 |

## 5. Conclusions

In this paper, we proposed a point–line-aware heterogeneous graph attention network for a visual SLAM system. Combining the point- and line-aware attention modules based on an attention-driven mechanism, the geometric association features of key regions was further extracted, and the model was simplified by a transfer-aware knowledge distillation strategy. By improving the accuracy of image point–line matching, a point–line heterogeneous graph attention network was proposed, which realized the feature aggregation by conducting learning on the intragraph and intergraph. Based on the optimal transport theory, we proposed a greedy inexact proximal point method that could effectively solve the point–line matching problem. Experiments on a public dataset and a self-made dataset showed qualitatively and quantitatively that our model had stronger robustness and a better generalization ability. One limitation of our feature matching was that it was not easy to estimate the pose error due to the interference of dynamic objects. Thus, in a future study, we will introduce the cross-frame semantic information in the network for dynamic environments.

**Author Contributions:** Conceptualization, Y.L. and H.S.; methodology, Y.L.; software, H.S.; validation, S.D.; formal analysis, H.S.; investigation, Y.L.; resources, H.S.; data curation, Y.L.; writing—original draft preparation, H.S.; writing—review and editing, Y.L.; visualization, H.S.; supervision, Y.L.; project administration, Y.L.; funding acquisition, Y.L. All authors have read and agreed to the published version of the manuscript.

**Funding:** This research was funded by NSFC 61972353, NSF IIS-1816511, OAC-1910469 and Strategic Cooperation Technology Proiects of CNPC and CUPB: ZLZX2020-05.

**Institutional Review Board Statement:** Not applicable.

**Informed Consent Statement:** Not applicable.

**Data Availability Statement:** Not applicable.

**Conflicts of Interest:** The authors declare no conflict of interest.

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
