# Peer review of "Point–Line-Aware Heterogeneous Graph Attention Network for Visual SLAM System"

_applsci, doi:10.3390/app13063816_

Round 1
Reviewer 1 Report
The paper proposes a smart new approach to visual Simultaneous Localization and Mapping based on a point-line aware heterogeneous graph attention network.
Comprehensive testing reveals the advantages of the new approach. The paper deserves great praise, since it is a step forward in visual SLAM.
Line Hints / Typos
135 ReLU activation -> please explain ReLU
148 element-wish summation -> element-wise summation
153 pixel value -> pixel values
192 are convolutions of size -> please, specify size
210 we further compresses -> compress
211 A aware -> An aware
239 Consider two image -> images
240 let d -> Let
241 we utilize -> We
250 We proposes -> propose
264 for nodes -> For
345 We tests -> test
371 we intercepts -> intercept
400 were given -> are given
401 is significantly improve -> significantly improves
Reviewer 2 Report
This manuscript entitled “Point-Line Aware Heterogeneous Graph Attention Network for Visual SLAM System", describes an original contribution. This paper presents an efficient visual SLAM system that uses a point-line aware heterogeneous graph attention network to address the challenges of estimating robot pose and reconstructing 3D maps in complex industrial scenarios. The system leverages geometric relationships between points and lines to extract features and improve accuracy. A knowledge distillation strategy of aware transfer optimizes the network model for real-time performance. A point-line heterogeneous graph attention network improves the accuracy of point-line matching, and a greedy strategy is used to solve the optimization problem. Experiments on KITTI and self-made datasets demonstrate superior effectiveness, accuracy, and adaptability compared to state-of-the-art methods in visual SLAM.
The paper proposes a new visual SLAM system based on a point-line aware heterogeneous graph attention network to address the challenge of estimating the robot's position and creating a 3D map of the surrounding environment. The system combines points and line segments to solve the problem of insufficient reliable features in traditional approaches. The proposed system consists of a simultaneous feature extraction network and a point-line heterogeneous graph attention network. The point-line aware attention module guides the network's focus on the features of both points and lines in images. The network model is optimized using a knowledge distillation strategy of aware transfer to improve the system's real-time performance. The point-line matching process is transformed into an optimal transport problem, and a greedy inexact proximal point method is used to solve the optimization problem. Experiments on public and self-made datasets demonstrate that the proposed method has superior effectiveness, accuracy, and adaptability to the state-of-the-art visual SLAM methods. The limitation of the proposed method is the difficulty in estimating the pose error in dynamic environments, which may be addressed in future studies by introducing cross-frame semantic information into the network.
The topic is very interesting and the manuscript is well-written and presents a significant contribution. Nevertheless, it requires clarification to improve this work for a potential publication in the MDPI Journal.
1. In this paper, the first question that needs to be addressed is: what is the problem that the paper aims to solve? To answer this question, the authors need to provide a clear and concise statement of the problem and its significance. They should also motivate why this problem is important and explain how their proposed approach is different from existing solutions in the literature. Additionally, it is important to demonstrate the utility and advantages of their approach, including any potential benefits over previous methods like Smooth Variable Structure Filter based-SLAM. By doing so, the authors can provide a compelling argument for the value of their research and its potential impact in the field of Simultaneous Localization And Mapping.
2. Explain in detail what is the proposed solution to the problem.
3. What are the dynamical models used for the quadruped robot?
4. What is the communication topology assumed to be?
5. How is the system organized?
6. It is necessary to change figures with a higher resolution.
7. What is the primary challenge addressed by the proposed visual SLAM system?
8. How does the proposed system combine points and line segments to overcome the problem of inadequate reliable features in traditional approaches?
9. What are the two main components of the proposed visual SLAM system?
10. How does the point-line aware attention module enhance the efficiency and accuracy of the network's features?
11. How is the network model optimized to improve the system's real-time performance?
12. What is the point-line matching process, and how is it transformed into an optimal transport problem?
13. What method is used to solve the optimization problem in the point-line matching process?
14. What types of datasets were used in the experiments to evaluate the proposed method?
15. How does the proposed method compare to state-of-the-art visual SLAM methods in terms of effectiveness, accuracy, and adaptability? You should add some papers which talk about SVSF-based SLAM.
16. What is the limitation of the proposed method?
17. How might future studies address the limitation of the proposed method?
18. What is the significance of estimating the robot's position and creating a 3D map of the surrounding environment in SLAM?
19. How does the proposed system contribute to the advancement of SLAM technology?
20. How might the proposed system be applied in real-world scenarios?
21. What are the potential benefits of using the proposed system over traditional SLAM approaches?
To attain acceptance, the paper necessitates thorough revisions with minor modifications. The authors should emphasize their contributions compared to previous related works, and mention recent challenges with their references, for example, SVSF-based SLAM. Overall, this paper is well-structured, and the results seem mathematically accurate, making it an intriguing paper with only minor corrections. Furthermore, the authors should carefully proofread the paper for any expression and grammar errors.
